# Beyond Belief: Exploring the Alignment of Self-Efficacy, Self-Prediction, Self-Perception, and Actual Performance Measurement in a Squat Jump Performance—A Pilot Study

**DOI:** 10.3390/jfmk9010016

**Published:** 2024-01-03

**Authors:** Alessandro Cudicio, Valeria Agosti

**Affiliations:** 1Department of Human and Social Sciences, University of Bergamo, 24129 Bergamo, Italy; 2Department of Humanities, Philosophy and Education, University of Salerno, 84084 Fisciano, Italy; vaagosti@unisa.it

**Keywords:** athletic performance, self-efficacy, self-prediction, self-perception, squat jump

## Abstract

It is widely accepted that athletic performance emerges from a complex interaction between physical and cognitive features. Several studies highlighted self-efficacy (SE) in the cognitive domain of athletic performance, but no studies have correlated SE with sport-specific tasks. According to Bandura, this study explored SE and its relationship with self-prediction (SP), self-perception (PSJ), and actual performance in a squat jump (SJ). Thirty-nine healthy collegiate students were assessed using an SE questionnaire, an SP measurement tool, and a validated optical system for actual SJ performance. An SE score and an SE esteem index (SEE) were determined. The alignment between an individual’s SP of their SJ performance and their SE beliefs was also examined. The data revealed a significant correlation between SE score and both SJ (r = 0.432; *p* = 0.006) and SP (r = 0.441; *p* = 0.005). Furthermore, disparities among the actual SJ, SP, and SEE were statistically non-significant, implying a congruence between self-belief and performance. With a deeper understanding of the interaction between SE, SP, and sport-specific tasks, sports professionals could develop targeted interventions to enhance athletes’ overall athletic achievements and apply SE as a feature linking physical and cognitive athletic performance.

## 1. Introduction

Athletic performance is the result of a perfect balance among physical, cognitive, technical, and tactical elements that allow the athlete to achieve successful outcomes [1,2,3]. In athletic performance and in sport activities, cognitive functions and skills were identified as features useful to recognize information from the environment and link this to our background knowledge to better plan, organize, and execute the appropriate motor behavior [4,5]. An especially interesting and important cognitive feature in organizing athletic and sport performance is self-efficacy (SE), which refers to an individual’s belief in their ability to effectively plan and carry out the necessary actions to achieve specific goals [6]. Indeed, SE is emerging as a psychological skill training tool to influence an athlete’s performance [7] and sustain and ameliorate sport-specific performance over time [8,9]. Over the years, several studies have shown a positive correlation between SE and sport performance where, across various sporting and athletic endeavors, higher SE tends to be associated with enhanced performance outcomes [10,11,12,13,14,15,16]. On the other hand, Moritz et al. [10] outlined the limitations and perspectives of this relationship contrary to Bandura’s idea of sport performance, referred to as an achievement that needs specific and objective (quantitative outcomes) descriptive criteria [6]. In this vein, for SE to be effective in athletic and sport performance, it should be considered as a qualitative skill that requires a quantitative outcome [6,10,17]. However, even if the quantitative component of the SE covers a significant importance in the realm of sport performance, it is also fundamental to not consider SE only as the actual performance (outcome) but rather as the judgment regarding what the individual can accomplish during the actual performance (goal). In this way, SE could be considered a cognitive feature useful to build subjective performance consistent with the environmental information and the sport-specific requirements [4,5,18]. To reduce discrepancies between goals and outcomes, Bandura outlined that it is crucial to introduce the athlete’s own objective self-prediction (SP) of performance in the assessment measure [6]. Furthermore, to ensure valid and reliable outcomes, SE should be assessed in a task-specific manner; more specifically, the measures of SE and performance should be concordant [10]. To ensure this concordance, we chose the squat jump (SJ) as a useful task-specific performance for our study because it operates as a straightforward and effective test unaffected by physiological confounders [19].

SJ is a technique of the vertical jump beginning from a static, semi-squatting position useful in harnessing and releasing the elastic energy stored within the musculotendon complex, which sheds light on leg power capabilities [20]. SJ is a complex motor skill that requires complex motor coordination involving multiple joints but also requires perfect organization of motor behavior between the upper and lower body; no cognitive abilities are required. The height achieved in the SJ, both for inactive individuals and elite athletes, is indicative of explosive muscle strength and correlates with performance attributes like speed, agility, and power [21].

We assumed that shedding light on the intricate interplay between SE, SP, and actual SJ performance, also using a perceived SJ height (PSJ), could have potential implications for optimizing training and coaching cognitive strategies in sport science. To the best of our knowledge, no study has yet explored the nature of the relationship between SE and sport performance in a task-specific movement.

Designed as a pilot study, the aim of our research was to objectively assess the nature of the relationship between SE and sport performance in a task-specific movement and to identify the role of SE as a cognitive feature in sport performance.

## 2. Materials and Methods

### 2.1. Participants

A total of 39 healthy college students (27 M; 12 F), without any ongoing or history of neurological, orthopedic, and cardiac or systemic diseases that could interfere with physical performance, voluntarily participated during their curricular educational activities. All participants were assessed by anthropometric measurements with an electronic scale 872TM and a stadiometer 214TM (SECA, Hamburg, Germany) and physical activity level [22,23,24,25] (see Table 1). To ensure standardization within the group, only physically active individuals were included in the research. For this purpose, we utilized the IPAQ results to exclude individuals with a score below 700 MET/min/week [26]. Data were collected in adherence to all privacy policy procedures, and written informed consent was obtained from all participants in accordance with the Declaration of Helsinki [27].

The sample size for Pearson’s correlation was determined using power analysis conducted in G-POWER (ver. 3.1.9.7, Düsseldorf, Germany) using an alpha of 0.05, a power of 0.90, and a medium effect size (rho = 0.5) for a two-tailed test. Based on the assumptions, the required minimum sample size was determined to be thirty-seven [28].

### 2.2. Experimental Procedure

Before starting, all participants were informed of the study’s objectives and procedures. Subsequently, all participants underwent an SE, SP, SJ, and PSJ assessment conducted as follows (study timeline shown in Figure 1):

SE was assessed, according to Bandura [29], by an 8-item questionnaire using a five-level Likert scale (Q1–Q8), which was individually administered through Google Forms by means of their personal devices. The questionnaire was split into two sections. The first provided a graphic depiction of the correct technique for executing an SJ, consisting of a 6-item five-level Likert scale (Q1–Q6) graded from “not at all confident-1” to “extremely confident-5”. This aimed to gauge the participants’ confidence in performing an SJ at a specific height and confidence in their answers (height ranges were sex-specific). The second consisted of a 2-item five-level Likert scale (Q7), graded from “not good at all (below the 20th percentile)-1” to “extremely good (above the 80th percentile)-5”, aimed to grade performance compared to their peers within the study group. The final item (Q8), a five-level Likert scale graded from “not at all confident-1” to “extremely confident-5”, aimed to inquire about the participants’ confidence in all their previous responses. To evaluate questionnaire results, two original SE scores were calculated, as explained in the subsequent statistical analysis section.

SP and PSJ were assessed by means of a custom-made vertical graduated rod, measured in centimeters (cm). SP, defined as the participants’ subjective expectancy in jump height measure, was assessed only before the first jump, asking each participant to predict the maximum height they could achieve in an SJ by marking it on. PSJ, defined as the participant’s self-perception in the height measure of the SJ performed, was assessed after each SJ trial, hence having experienced the execution, asking each participant to evaluate the perceived SJ height achieved in each performance. For both measurements, participants were asked to physically mark their predicted and perceived SJ height on the vertical graduated rod, illustrated in Figure 2, by simply pointing at it, transforming their SP and PSJ in a measurable (cm) outcome. To ensure both unbiased data collection and an uninfluenced response by external feedback, only the investigator had access to the gradations on the rod.

SJ performance was measured by means of a validated optical system (Optojump Microgate, Bolzano, Italy) [30]. After a warm-up that did not involve SJ, each participant completed three SJ trials, observing a rest period of 5 min between each trial to prevent fatigue from impacting muscle activation [31]. The protocol for executing the SJ was carried out following standardized procedures [32]. The height of each SJ trial and the average SJ height were used for the subsequent statistical analysis.

### 2.3. Statistical Analysis 

Descriptive and inferential statistical analyses were conducted using the Jamovi for Windows statistical package (ver. 2.3.23, Sydney, Australia). A type I error rate of 0.05 to establish statistical significance was applied.

The Shapiro–Wilk test was employed to assess the normality of the distributions of key variables. The test indicated that the distribution of weight and BMI does not significantly deviate from normality (*p* = 0.288 and *p* = 0.090, respectively). In contrast, height showed a slight but significant deviation from normality (*p* = 0.046). Age and IPAQ scores both exhibited significant departures from a normal distribution (*p* < 0.001 for both). Data are shown in Table 1. We applied parametric hypothesis tests even if the population is not normally distributed. Indeed, it was established that if the sample size is large enough (*n* = 30), the central limit theorem comes into effect and creates sampling distributions that are close to normal [33]. 

To evaluate the internal consistency of the SE questionnaire, Cronbach’s α and McDonald’s ω were assessed.

Before conducting repeated measures ANOVA and Pearson’s statistical analysis, we calculate two SE values and two values showing the discrepancies between SP and SJ and between PSJ and SJ:

(1) The SE questionnaire score (SE score) was calculated to obtain a unique and comprehensive individual SE value. We calculated the value employing the following expression:SE score = (((*t* − (Q1 + Q2 + Q3 + Q4 + Q5) + *r* − Q6 + *r* − Q7 + *r* − Q8 + ∣Q6 − Q8∣) − *y*)/*y*) × 100(1)
where (*t*) is a constant representing the maximum obtainable value; (*r*) is a constant corresponding to the maximum value of an answer; and Q1 to Q5 are questions related to an individual’s confidence in performing a specific height jump; Q6 reflects the respondent’s confidence in their answers from Q1 to Q5; Q7 assesses their self-evaluation compared to peers in the study group (self-reported percentile); Q8 pertains to their overall confidence in their previous responses; the absolute value of the difference between Q6 and Q8 represents the distance between the respondent’s confidence in their answers from Q1 to Q5 and their overall confidence in their previous responses; (*y*) represents the highest achievable value. According to this equation, a higher value obtained in the SE score corresponds to subjects with higher SE.

(2) The SE esteem measure (SEE) was calculated to esteem the SJ height based on participants’ confidence in performing an SJ at a specific height. We calculated the value employing the following expression:SEE = (Q1 + Q2 + Q3 + Q4 + Q5) × (*c*/*r*),(2)
where (*c*) is a constant denoting the maximum value within the range defined by the answer; (*r*) is a constant corresponding to the maximum value of an answer; and Q1 to Q5 are questions related to an individual’s confidence in performing a specific height jump.

SEE allows the questionnaire data to be switched to measurable data in cm. A higher SEE corresponds to subjects with a higher SE jump-specific belief expressed in cm.

(3) The delta total error (DTE) was calculated to identify the error entity between SP and SJ or between PSJ and SJ. We calculated the value employing the following expression:DTE = (SP or PSJ) − SJ,(3)

(4) The delta absolute error (DAE) was calculated to identify the absolute error entity between SP and SJ or between PSJ and SJ. We calculated the value employing the following expression:DAE = ∣(SP or PSJ) − SJ∣,(4)

The equations and delta values were used in the statistical analysis as follows.

A repeated measures ANOVA was performed to evaluate variations in SJ, SEE, and SP and variations in SP and PSJ errors among repetitions. 

Pearson’s *r* correlation was performed to investigate the associations between the SE score and both SP and actual SJ height, between SEE and SJ, between SEE and SP, and to compare the correlation between the self-reported percentile (Q7) and the SJ percentile.

## 3. Results

The 8-item SE questionnaire showed sufficient internal consistency and fit (Cronbach’s α = 0.885; McDonald’s ω = 0.895) for all the items.

The group results of the tests and questionnaires are reported in Table 2. 

ANOVA repeated measure analysis showed no statistical differences (*p* = 0.666; F (2.76) = 0.409) in the measured values for SJ height, SP, and SEE, as illustrated in Figure 3. 

ANOVA repeated measure analysis showed no significant changes in DTE across different time points (*p* = 0.869; F (3.114) = 0.239), as depicted in Figure 4. The DTE measurements after each SJ exhibited a consistent trend, showing a decrease from the pre-assessment (PRE) to the first post-assessment (POST 1), further declining to the second post-assessment (POST 2), and ultimately stabilizing at the last post-assessment (POST 3). Moreover, when examining DAE values (*p* = 0.075; F (3.114) = 2.36), illustrated in Figure 5, a similar pattern emerged. DAE decreased from PRE to POST 1, followed by a further decline to POST 2, and eventually reaching the lowest point at POST 3. DTE and DAE means are reported in Table 3.

The coefficient of variation of DAE exhibited a gradual and consistent decrease across the four evaluations conducted (0.82, 0.71, 0.78, and 0.68). 

Pearson’s analysis showed a positive correlation between the self-reported percentile (Q7) and SJ percentile (r = 0.427; *p* = 0.007). The SE score exhibited significant correlations with both the SJ (r = 0.432; *p* = 0.006) (see Figure 6) and the SP (r = 0.441; *p* = 0.005) (see Figure 7). The SEE exhibited significant correlations with both the SJ (r = 0.440; *p* = 0.005) and the SP (r = 0.463; *p* = 0.002).

## 4. Discussion

As a pilot study, we presented preliminary findings in studying SE as a cognitive feature of sport performance. We started from Bandura’s assumptions defining sport performance as “an objective outcome that must be specified by precise and objective descriptive indicators and measures that are generally quantitative and typically more based on results” and SE in sport performance as “a performer’s belief that he or she can execute a behavior required to produce a certain outcome successfully” [6]. Due to the quantitative nature of sport performance, SE in sport fields must be contextualized in coherence and in its relationship with specific skills or tasks [10]. In this line, our study did not aim to investigate the trainability of SE or SJ but to objectively assess the nature of the relationship between SE and sport performance in a task-specific movement skill using original measures, scores, and procedures. 

SJ is a fundamental athletic skill employed across a wide array of sports ranging from individual disciplines to team sports [34]. SJ has been widely shown to be an important tool in evaluating lower limb power, peripheral fatigue, elastic component compliance, and biomechanical domains to improve athletic performance. In our study, we used SJ measurements as an outcome to compare SE, SP, and PSJ. 

SE is a cognitive feature widely investigated in sport and training contexts. However, previous studies highlighted the need for sports research employing the SE construct in relation to specific self-expectancy outcomes [35,36,37,38]. Previous studies also highlighted the need, in this area of the literature, for a univocal nomenclature required in assuming not only terminologies but also in defining original indices, measures, and procedures [39]. 

From these needs, as highlighted in previous studies, in our study, we explored SE using original scores and procedures in data collection. In detail, SP, PSJ, SE score, and SEE are values that we linked with actual performance in an SJ in order to better investigate the link between the motor task and the SE of the motor task. Furthermore, to obtain these values, we investigated the subjective perception of the motor task using two original methods of data collection: the first was organizing an 8-item five-level Likert scale SE questionnaire. We managed the questionnaire in two sections: one aimed to gauge the participants’ confidence in performing an SJ at a specific height and confidence in their answers, the subsequent aimed to grade performance compared to their peers within the study group and also to inquire about the participants’ confidence in all their previous responses; the second, using a custom-made vertical graduated rod transforming subjective SP and PSJ in a measurable (cm) outcome. This method, providing a tangible and standardized measure, we hypothesized that it might be useful to allow for a more in-depth understanding of participants’ subjective experiences during the SJ performance and also ensure consistency in the participants’ responses. This direct involvement of participants not only added a qualitative dimension to the data but also allowed for a more in-depth understanding of their subjective experiences during the jumping task. The use of a vertical graduated rod provided a tangible and standardized measure, ensuring consistency in the participants’ responses. This detailed method not only captured the participants’ objective performance but also delved into their perceptions, offering valuable insights into their self-assessment and confidence levels during the jumping trials.

An intriguing initial finding is that the SJ height, SP, and SEE results have shown an interesting interplay. As can be seen in Figure 3, these three parameters exhibited minimal differences. Contrary to expectations, these disparities were revealed to be not statistically significant, highlighting a compelling aspect of the study’s findings (*p* = 0.666; F (2,76) = 0.409). This result suggested that participants’ SJ performance was remarkably aligned with both their SEE and the SP. This congruence indicated a fascinating harmony between participants’ beliefs in their abilities (as reflected in SEE) and the objective measurements of their performance. Such coherence between perception and reality not only underscores the participants’ accurate self-assessment but also raises intriguing questions about the psychological factors influencing their confidence levels and performance outcomes. Moreover, this outcome supports the reliability of the method employed in this and in past [40] experimental setups. Then, it could serve as an encouraging revelation to include the evaluation of SE and SP during training, thereby enhancing athletes’ self-awareness of their skills and, consequently, their overall performance.

Then, this study explored the alignment between individuals’ DTA and DAE throughout multiple repetitions, objectively measured by an optical system, of an SJ execution. The results of the study did not uncover any significant disparities in DTE or DAE values over the various sampling times. However, an intriguing trend emerged as the coefficient of variation of DAE values decreased from the initial assessment to the final one. This decline in error suggests that with repeated attempts, individuals may enhance the accuracy of their SPJ assessments, indicating a potential alignment between SPJ and SE that develops over time. It is conceivable that the extended practice of SPJ over time could contribute to athletes improving their cognitive perception of their performance. Moreover, as outlined in previous studies [41,42], implementing a specialized training protocol like goal-plus-feedback has been proven effective in enhancing SE. This highlights the importance of urging trainers and professionals in sport and motor sciences to adopt tailored training practices from an educational point of view [43,44]. Emphasizing the significance of dedicated attention to improving SE can significantly enhance the overall training experience and outcomes. There are new needs in sport and skill learning directly linked with post-cognitive approaches [45]. The study of SE thus understood, which transcends the boundaries of a mechanistic view of the human organism and its movement organization and aligns with complex systems theory [46,47], allows us to view athletic and sport performance as an emergent property from the complex interaction between organism and a variable environment [48,49]. Therefore, the human body plays a central role in the realm of movement, and the individual is not merely a means of motion but rather the very essence of the movement itself. In this perspective, the person is not just a passive entity but an active protagonist, shaping and driving the entire process of physical activity. As highlighted by Balague and colleagues [50], “The emergence of self-organizing coordinated movement patterns occurring at all scales and in all types of situations in sport dilutes the extant boundaries between technique and tactics”.

Furthermore, participants demonstrated the capability to gauge their performance level within the group. This observation remarked that participants’ self-assessments were in line with their actual SJ performance, suggesting that they had a realistic understanding of their capabilities compared to peers. This aligns with the observations made by Bandura [6] and Moritz [10], who emphasize the group’s impact on SE assessment. This factor could have a significant role in shaping specific training regimens. Athletes looking to enhance their SE might find it more advantageous to engage in group training or regular competitive activities rather than pursuing solitary or isolated training methods.

The significant correlations observed between the SE score (as well as SEE) and both the SJ height and SP height indicate that SE plays a role in an individual’s SJ performance. Although, in some cases, these interplays could be considered not extremely strong in terms of correlation strength, *p*-value < 0.01 and sample power stronger than 90% suggests that the found relationships are statistically significant and could be considered robust. This finding aligns with previous research that has demonstrated the influence of SE on various aspects of performance and achievement [10,11,16]. Athletes with higher SE tend to set more ambitious goals and exhibit greater determination to persist through rigorous training regimens. This confidence in their abilities translates into tangible improvements in athletic performance, as athletes approach competitions with a winning mindset and a belief in their capacity to excel [51,52,53]. The present study adds to this body of knowledge by highlighting the association between SE, SP, and SJ performance, providing further evidence of the transformative potential of SE in sport settings. The observed correlations between SE and SJ performance suggest that SE beliefs may serve as a predictor of an individual’s jump height. To our opinion, this could play a crucial role in sport environments, especially in competitive and situational sports. This finding is consistent with Bandura’s SE theory, which posits that individuals with stronger SE beliefs are more likely to engage in activities that lead to successful outcomes [10,11]. Therefore, assessing and enhancing SE beliefs could be a valuable strategy for coaches and sport professionals to optimize training programs and promote better performance outcomes.

The limitations of this study should be acknowledged. The study sample consisted of healthy young adults who were sport science students, which may limit the generalizability of the findings to other populations. Future research could include diverse samples with different age groups and athletic backgrounds to further explore the relationship between SE, SP, SPJ, and physical performance. Future studies could consider incorporating objective measures or observational assessments to complement self-report data. Additionally, the study focused on the SJ as the specific movement of interest. Including other jump techniques or different movements or skills could offer a more comprehensive and nuanced understanding of the subject matter. 

However, to our knowledge, this is the first study exploring the intricate interplay between a specific movement, such as SJ, and individuals’ SE, SP, and PSJ throughout multiple repetitions of SJ execution. Far from the aim to investigate the trainability of SE or SJ, our study aimed to respond to previous studies’ requests to objectively assess the nature of the relationship between SE and sport performance in a task-specific movement skill using original measures, scores, and procedures. 

## 5. Conclusions

As a pilot study, our research was conducted to test the practicality of methods and procedures for future application in larger-scale studies and to identify potential effects and associations that merit further exploration in a more extensive subsequent study [54]. SE was investigated as a cognitive feature useful to recognize information from the environment and link this to our background knowledge to better plan, organize, and execute the appropriate motor behavior.

In response to evolving requirements in sports and skill acquisition, it is essential to shift our focus to measuring and analyzing variables. Traditional methods often concentrate solely on either the individual or the external environment [55]. However, a more holistic approach is needed, one that considers the interaction between the athlete and their environment. Such a research approach aligns with the principles of post-cognitive theories [45]. These theories emphasize the importance of self-organization in addressing the challenges and problems encountered in performance tasks. Incorporating these concepts into sports training programs can lead to more effective learning designs that enhance the cognitive abilities of athletes.

Improving the knowledge of athletes’ SE could hold the potential to greatly benefit training and coaching strategies, ultimately leading to improved performance outcomes. Additionally, the study’s findings indicate that SP may benefit from repeated assessments and further training sessions, as evidenced by a decreasing coefficient of variation in error among the analyzed group. By gaining a deeper understanding of the relationship between SE, SP, PSJ, and physical performance, coaches and sport professionals could develop targeted interventions to enhance athletes’ overall athletic achievements and also apply SE as a feature linking physical and cognitive athletic performance.

## Figures and Tables

**Figure 1 jfmk-09-00016-f001:**
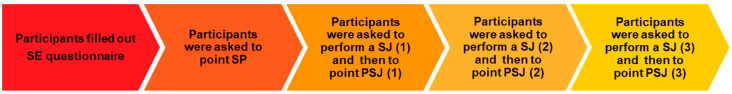
Study timeline. Self-efficacy (SE); self-prediction (SP); squat jump (SJ); perceived SJ (PSJ).

**Figure 2 jfmk-09-00016-f002:**
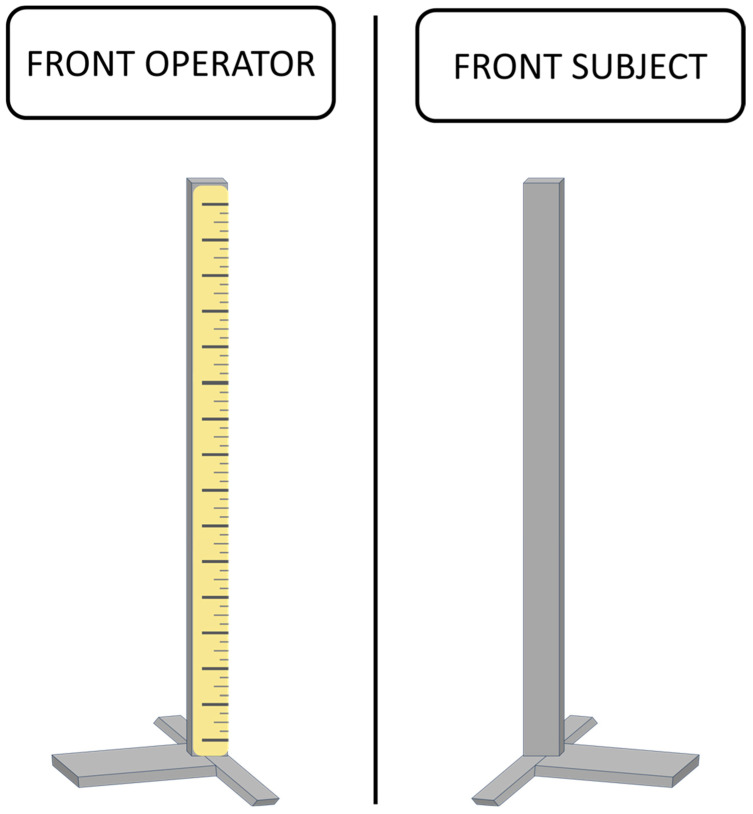
Graphical representation of the rod used to evaluate self-prediction and self-perception of the squat jump. On the left, the rod has been graduated to allow the operator to take the measurement; on the right, the smooth side of the rod shown to the subject during the evaluation.

**Figure 3 jfmk-09-00016-f003:**
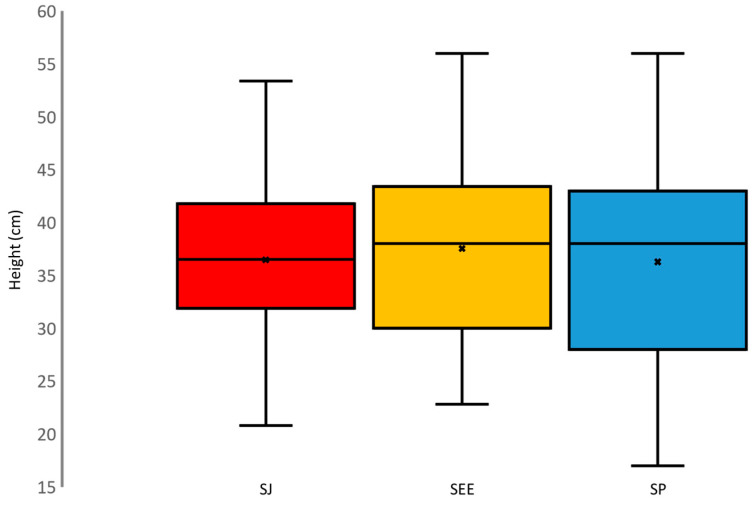
Boxplots of the distribution of SJ height measurements obtained during the SJ (in red), SSE (in yellow), and SP (in blue).

**Figure 4 jfmk-09-00016-f004:**
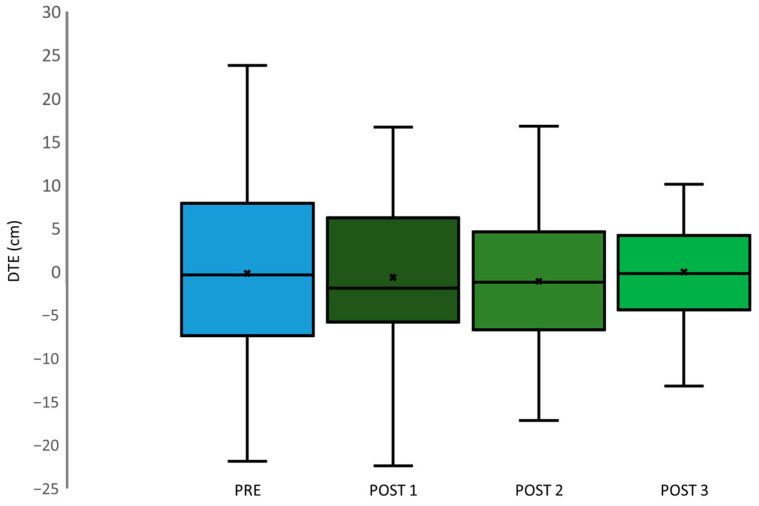
Boxplots depict distribution of DTE, PRE (light blue), POST 1 (dark green), POST 2 (green), and POST 3 (light green).

**Figure 5 jfmk-09-00016-f005:**
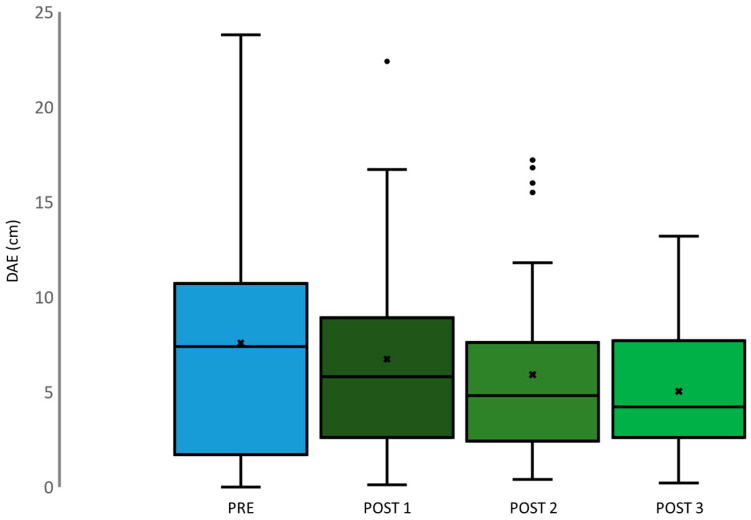
Boxplots depict distribution of DAE, PRE (light blue), POST 1 (dark green), POST 2 (green), and POST 3 (light green).

**Figure 6 jfmk-09-00016-f006:**
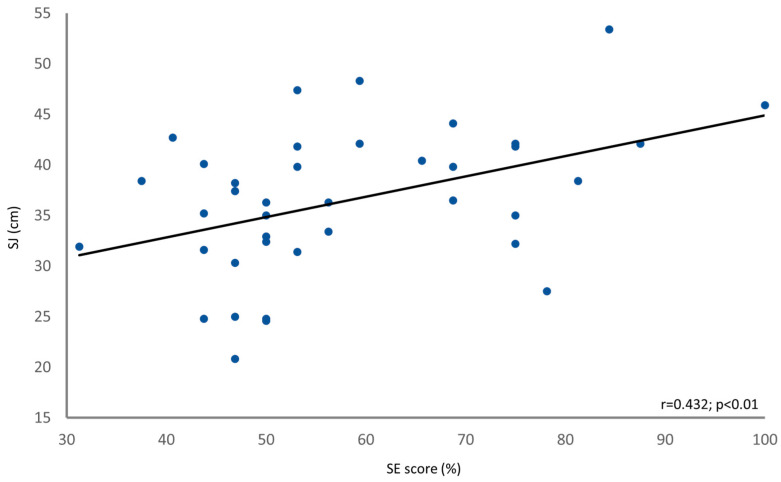
Linear correlation (solid black line) between the SE score (x-axis) and the SJ (y-axis). Each data point represents an individual subject.

**Figure 7 jfmk-09-00016-f007:**
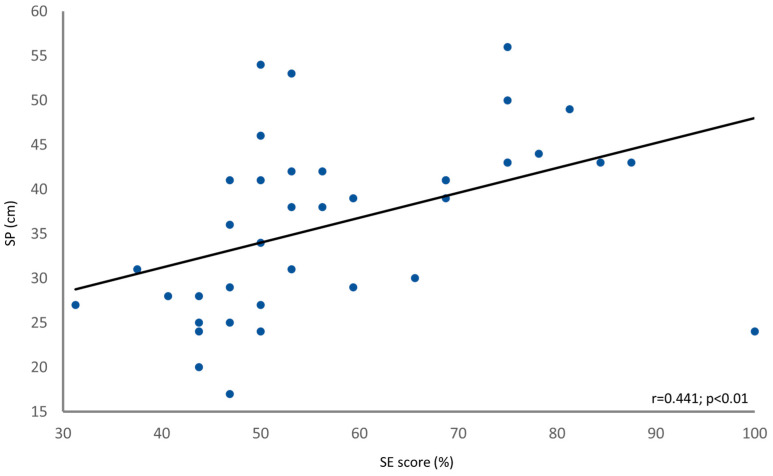
Linear correlation (solid black line) between the SE score (x-axis) and the SP (y-axis). Each data point represents an individual subject.

**Table 1 jfmk-09-00016-t001:** Demographic and descriptive data (yrs: years; m: male; f: female; kg: kilogram; cm: centimeters; BMI: body mass index; IPAQ: International Physical Activity Questionnaire; MET-min/wk: metabolic equivalent task minutes per week; SD: standard deviation).

Subjects Number	Sex (f/m)	Age (yrs) (Mean ±SD)	Weight (kg) (Mean ± SD)	Height (cm) (Mean ± SD)	BMI (Mean ± SD)	IPAQ (MET-min/wk) (Mean ± SD)
39	12/27	21.7 ± 1.8	70.2 ± 11.2	176 ± 9	22.6 ± 2.7	3797 ± 1979

**Table 2 jfmk-09-00016-t002:** Mean value and standard deviation for squat jump (SJ), self-efficacy score (SE score), self-efficacy esteem (SEE), self-prediction (SP), self-reported percentile (Q7), squat jump percentile (SJ percentile), and perceived squat jump (PSJ).

SJ (cm)	SE Score (%)	SEE (cm)	SP (cm)	Q7 (a.u.)	SJ Percentile (a.u.)	PSJ (cm)
36.5 ± 7.23	58.1 ± 10.55	37.5 ± 8.58	36.3 ± 9.7	3.1 ± 0.72	49.34 ± 29.8	36.28 ± 9.87

**Table 3 jfmk-09-00016-t003:** Mean value and standard deviation for delta total error (DTE), delta absolute error (DAE), and pre-assessment (PRE) and after the first (POST 1), the second (POST 2), and the third (POST 3) squat jump.

DTE PRE (cm)	DTE POST 1 (cm)	DTE POST 2 (cm)	DTE POST 3 (cm)	DAE PRE (cm)	DAE POST 1 (cm)	DAE POST 2 (cm)	DAE POST 3 (cm)
−0.2 ± 9.89	−0.6 ± 8.28	−1.1 ± 7.47	−0.1 ± 6.12	7.6 ± 6.24	6.7 ± 4.76	5.9 ± 4.61	5.1 ± 3.40

## Data Availability

The authors strongly encourage researchers with a vested interest to establish contact, as they are more than willing to share the data’s content upon request.

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
