# Peer review of "Beyond Belief: Exploring the Alignment of Self-Efficacy, Self-Prediction, Self-Perception, and Actual Performance Measurement in a Squat Jump Performance—A Pilot Study"

_jfmk, 2024, doi:10.3390/jfmk9010016_

Round 1

Reviewer 1 Report

Comments and Suggestions for Authors

This study examined the alignment of Self-Efficacy, Self-prediction, Self-perception, and actual performance measurement in a squat jump performance. Unfortunately, this study's rationale is inadequate. Furthermore, the experiment's internal and external validity is not adequate. Finally, statistical analysis and the interpretation of the results were not conducted in a scientific manner. Therefore, I believe this manuscript is not suitable for publication. More specific comments are the following:

Abstract:

1.     Lines 17-18: Authors should report the correlation size, not only the statistical significance. This is a moderate to low correlation. Statistical significance depends on the sample size.

Introduction:

1.     I’m not convinced that SJ is the right choice here. Previous research did show the importance of SE and SEE; however, SJ is a task highly dependent on leg muscles' explosive power. No skill is needed, nor is there any cognitive effort to perform it. Therefore, I believe you should have chosen a specific task with more cognitive impact. Otherwise, this relatively low correlation on a small sample size cannot be generalized.

2.     Moreover, the authors did not elaborate in the introduction on the second part of the experiment regarding the feedback and the use of vertical graduated rod. This seems to be two separate experiments poorly merged.

Methods:

1.     As I mentioned previously, this sample size is inadequate for a broader generalization of the results. The authors did not mention previous sports engagement. Students who were once engaged in sports may have better SE and SJ (since they trained for both), while students without sports backgrounds have low SJ and SE.

2.     Furthermore, the authors must elaborate on how they got this sample size (G*power?)

3.     Women and men are different in SJ, and I assume SE? Therefore, I believe they shouldn’t be in the one sample

4.     Figure 1 Illustration is wrong. Or do participants really jump this way? Arms should go up and in front when jumping, not back. Actually, for SJ protocol, arms should be on the hips.

5.     the authors did not check the data normality in the statistical analysis.

6.     Also, the SE score is expressed as a %. You cannot perform the parametric statistics on %. Data needs to be transformed. I would suggest asking a statistician for help.

7.     Finally, the authors did not present how the correlation coefficients were reported.

Results:

1.     Authors reported doing repeated ANOVA in methods; however, no results were presented.

2.     I’ve already mentioned in the abstract the size of the correlation coefficients should be reported, not only statistical significance. In all cases, r is around 0.45, resulting in R around 0.2, which means only 20% of the variance can be explained. The authors presented results and obtained correlations much higher and more important than they really are.

Discussion and Conclusions:

1. Based on my previous comments, a new approach to this problem is needed with new statistical analyses and interpretation of the results. Therefore, the Discussion and Conclusions also need rewriting.

Reviewer 2 Report

Comments and Suggestions for Authors

My recommendations are the following:

In the abstract, I recommend mentioning the number of subjects, not the cohort. As well as the applied tests and questionnaires.

Line 85 I recommend clarifying the following values - (IPAQ) (3797 ± 1979 MET/minutes per week)?? My recommendation is to mention an average value and a standard deviation, also to calculate Cronhback's α for the entire questionnaire.

I recommend deleting line 85-87, it is sufficient if it is mentioned that the subjects participated voluntarily.

Lines 93-105, it is enough to mention once that a five-level Likert-Scale was used, I recommend the correction.

Lines 329-330 recommend deletion, this conclusion is too general and unfocused.

I recommend rewriting the Conclusions section, focused on the study results. They are too general and evasive. The study finds some results at a given moment, you do not have any methodological intervention, in this sense the strategies you talk about are not relevant.

Reviewer 3 Report

Comments and Suggestions for Authors

Overall, the procedures of the study are good but the design need to be improved. Besides, the idea is not very original. Below there are some issues that need to be addressed:

Abstract

- Reduce the extension of the initial background. You can add context in the introduction.

- Line 15. Be more specific about the technological measure mentioned

- Avoid personal forms such as: "our" or "we"

Introduction

- The information included in the first paragraphs is very general and sometimes not necessary. . Please, reduce the content and focus on the relevant evidence/findings from prior research that is associated with your investigation

- The research has no hypothesis. Please, include the hypothesis of the research at the end of the introduction.

- Before the aims of the study, a mention to why your investigation is importance and necessary should be included. The previous paragraph is too long and little messy. It should response to: Why your study is necessary?

Materials and Methods

- Please, explain the inclusion and exclusion criteria of the participants of the study. Include age, academic level and other general information to see the homogeneity of participants and check if they can be considered a sample of population.

- The explanation of the procedures of the SJ are a little messy. Please, check previous authors. Perhaps, it is recommended to only indicate something like "SJ was performed following the procedures defined by XXXX"

- Table 1 seems unecessary. Please define the sample in the text and delete the table

Results

- It looks loke table 2 and figure 2 are displaying the same information. If so, please only indicate it in the figure otherwise it is redundant.

- Same for the rest of the tables and figures. Do not duplicate the information

Discussion

- In the first paragraph the key results must be better highlighted and indicate whether the hypothesis was supported.

- Overall, the parts with general statements are too long. Include more specific references to previous studies (even in other strength or power test) and their findings. Are they similar? Why do you think is that?

- The correlations between SE and SJ are too small to suggest that SE can be considered and predictor. 0.4 is not a strong interrelationship. Please treat the results with caution and rewrite that part (line 331 to 347).

- Include what are the implications of the results of your research. How does this study may help to improve things in the future? Why are they important to the field?

Conclusions

- Please be more clear about the results of your study here.

- Paragraph from lines 372 to 377 should be placed in the discussion

- Delete last paragraph or include it in the discussion

Reviewer 4 Report

Comments and Suggestions for Authors

Dear Authors

You have written an interesting paper focusing on objectively assessing the nature of the relationship between Self-Efficacy and performance in a task-specific movement and identifying the role of Self-Efficacy as a cognitive feature in sports performance.

However, some parts need to be addressed for greater clarity_

The introduction leads well to the main study rationale.

Methods: Report the activity levels of participants (also sporting experience) and their experience in these movements as this can have a detrimental effect on SE.

Report how you measured body weight and body height.

Did you follow Helsinki Declaration and did they sign a written consent form? report

Figure 1 - explain abbreviations

Were they any practice trials before demonstrations of jumps? report

Add a picture of the ''custom made vertical graduated rod''

What was the break between jumps and jump tasks? report

Results are well presented

The discussion is solid and connects to relevant references.

However, the limitations section should acknowledge the lack of info of participants' sports and training backgrounds as some may have more experience in these tasks and could know their performance in these tasks. So the sample is not ideal

The conclusion is solid.

Overall, I recommend a major revision.

Kind regards

Comments on the Quality of English Language

Moderate editing of the English language required

Round 2

Reviewer 1 Report

Comments and Suggestions for Authors

Dear authors,

Unfortunately, the answers and changes you provided do not improve this manuscript sufficiently. Instead of focusing on making some important and necessary changes to improve its quality, your focus was to prove me wrong. Therefore, I’m sticking with my decision that this paper should be rejected for publication.

Reviewer 4 Report

Comments and Suggestions for Authors

Dear Authors,

Thank you for addressing my questions and suggestions. The paper's quality improved. However, I can not agree with the response to question 7 - a picture of your custom-made vertical graduated rod. Your response is not valid, especially in a journal without any limitation in this regard.

Please tell me how can someone replicate your study in this sense if we don't know the exact settings and no one has seen this setting device. So without this evidence-picture, your research is not fully replicable. Therefore, add this picture.

Kind regards

Comments on the Quality of English Language

Minor editing of the English language required

Author Response

We acknowledge the reviewer's feedback and have incorporated it by enhancing the manuscript with a graphical representation illustrating the rod's characteristics.

Round 3

Reviewer 4 Report

Comments and Suggestions for Authors

Dear Authors

Thank you for adding the last part of the manuscript. In my opinion, the paper is suitable for publication.

Kind regards